# Structural basis of substrate recognition and translocation by human ABCA4

Tian Xie [1,2], Zike Zhang[1,2], Qi Fang[1], Bowen Du[1] & Xin Gong [1✉]

Human ATP-binding cassette (ABC) subfamily A (ABCA) transporters mediate the transport of various lipid compounds across the membrane. Mutations in human ABCA transporters have been described to cause severe hereditary disorders associated with impaired lipid transport. However, little is known about the mechanistic details of substrate recognition and translocation by ABCA transporters. Here, we present three cryo-EM structures of human ABCA4, a retina-specific ABCA transporter, in distinct functional states at resolutions of 3.3–3.4 Å. In the nucleotide-free state, the two transmembrane domains (TMDs) exhibit a lateral-opening conformation, allowing the lateral entry of substrate from the lipid bilayer. The N-retinylidene-phosphatidylethanolamine (NRPE), the physiological lipid substrate of ABCA4, is sandwiched between the two TMDs in the luminal leaflet and is further stabilized by an extended loop from extracellular domain 1. In the ATP-bound state, the two TMDs display a closed conformation, which precludes the substrate binding. Our study provides a molecular basis to understand the mechanism of ABCA4-mediated NRPE recognition and translocation, and suggests a common 'lateral access and extrusion' mechanism for ABCA-mediated lipid transport.

[1] Department of Biology, Southern University of Science and Technology, Shenzhen, Guangdong, China. [2] These authors contributed equally: Tian Xie, Zike Zhang. ✉email: gongx@sustech.edu.cn

Human ABC transporter superfamily, consisting of 48 members arranged in seven subfamilies (A to G), can transport a broad spectrum of substrates across the plasma or intercellular membranes utilizing the energy of ATP hydrolysis[1–3]. The structure of a full ABC transporter shares a common core architecture, comprised of two transmembrane domains (TMDs) that bind substrates and two nucleotide-binding domains (NBDs) that hydrolyze ATP[4]. The ABCA subfamily consists of 12 full transporters and can be distinguished from other ABC transporters by the presence of a large extracellular domain (ECD) between the first and second TM segments of each TMD and a cytoplasmic regulatory domain (RD) downstream of each NBD[5–7].

The ABCA transporters are responsible for the transport of a variety of lipid or lipid-based substrates in different organs and cell types[8–10]. Mutations in several ABCA members have been linked to severe human inherited diseases related with impaired lipid transport, such as Tangier disease and familial high-density lipoprotein (HDL) deficiency caused by mutations in ABCA1 resulting in reduced cholesterol and phospholipid efflux from cells and a decrease in circulating HDL[11–14]; fatal surfactant deficiency and pediatric interstitial lung disease caused by mutations in ABCA3 linked to impaired surfactant lipid secretion in lung cells[15,16]; and lamellar and harlequin ichthyosis caused by mutations in ABCA12 resulting in defective lipid secretion in keratinocytes that leads to skin lipid barrier dysfunction[17–19]. The structure of human ABCA1 resolved in substrate- and nucleotide-free state suggests a plausible "lateral access" mechanism for ABCA1-mediated lipid export[20]. To understand the detailed mechanism of lipid recognition and translocation by ABCA transporters, more structures of ABCA transporters in complex with their substrates and in distinct conformations would be required.

ABCA4 (also known as ABCR or the rim protein) is a retinal-specific member of the ABCA subfamily predominantly expressed in the outer segment disk membranes of photoreceptor cells[21–23], and was the first ABCA transporter that has been linked to genetic disease[24]. The *ABCA4* gene, first cloned and characterized in 1997, is related to Stargardt disease, the most common heritable macular degenerative disorder[24], and implicated in several other severe visual disorders, such as age-related macular degeneration[25], retinitis pigmentosa[26] and cone-rod dystrophy[27]. Available evidence suggested that ABCA4 may act as a flippase to translocate N-retinylidene-phosphatidylethanolamine (NRPE), a reversible covalent adduct of all-trans retinal (ATR) and phosphatidylethanolamine (PE), from the luminal leaflet to the cytoplasmic leaflet of the disk membrane[28–31]. After translocation, NRPE can be hydrolyzed to ATR and PE, thereby enabling ATR to be reduced to all-trans retinol (ATRol) by the cytoplasmic retinal dehydrogenase (RDH)[32]. Through this proposed model, the toxic ATR in photoreceptor cells following photoexcitation is removed from the luminal leaflet of the disc membrane and then goes into the retinoid cycle for regeneration of 11-cis-retinal[33,34]. Otherwise, ATR and NRPE would accumulate in the luminal membrane leaflet and form toxic bisretinoid derivatives that may cause retinal degeneration[35–37]. Apart from NRPE, PE has also been suggested to be a substrate of ABCA4[30,38]. However, the molecular basis for the substrate recognition by ABCA4 requires further investigation.

Here, we show the molecular structures of human ABCA4 determined in three distinct functional states by single-particle cryo-EM to 3.3–3.4 Å. These structures delineate the conformational transitions of ABCA4 to flip NRPE and suggest a "lateral access and extrusion" mechanism for ABCA4-mediated NRPE recognition and translocation, a mechanism that might also be adopted by other ABCA transporters.

## Results

### Biochemical characterization and structure determination of human ABCA4.

The full-length human ABCA4 (hABCA4), comprised of 2273 amino acids, was recombinantly expressed in human embryonic kidney (HEK) cells and purified to homogeneity in detergent micelles (Supplementary Fig. 1a). The ABCA4 protein was extracted from the membrane with n-dodecyl-β-D-maltoside (DDM) plus cholesteryl hemi-succinate (CHS) and exchanged to glyco-diosgenin (GDN) detergent during purification. The purified wild-type (WT) ABCA4 protein displayed robust ATPase activity with $K_m$ and $V_{max}$ values of 0.11 mM and 107.7 nmol/min/mg, respectively (Supplementary Fig. 1b). As a negative control, a variant of ABCA4 generated by replacing both catalytic glutamates in Walker B motifs of NBD1 and NBD2 with glutamines (E1087Q/E2096Q, named ABCA4$_{QQ}$) exhibited completely abolished ATP hydrolysis (Supplementary Fig. 1b).

It was reported that the ATPase activity of CHAPS solubilized and purified ABCA4 depended on the presence of phospholipids during protein preparation[39]. The robust ATPase activity of our purified ABCA4 protein indicates the presence of endogenous phospholipids co-purified with ABCA4. Consistent with this, phosphatidylethanolamine (PE) and phosphatidylcholine (PC) were identified from the purified ABCA4 by both thin-layer chromatography (TLC) and mass spectrometry methods (Supplementary Fig. 1c, d).

To gain structural insights into ABCA4, we first determined the structure of WT hABCA4 in the absence of NRPE and nucleotide (apo ABCA4) by single particle cryo-EM (Supplementary Fig. 2). The overall resolution of the structure is estimated to be 3.3 Å, allowing de novo model building of ABCA4 for most of the protein (Supplementary Fig. 3). In total, 2007 residues were built with 1815 side chains assigned (Supplementary Table 1). Since the electron density for lid is relatively poor, the model building was accomplished based on the homology model of lid from ABCA1. In addition to the polypeptide chain, 17 sugar moieties were built into 9 glycosylation sites on ECD, which further validated the sequence assignment. The general architecture of the structure is quite similar to the torch-shaped hABCA1 structure in the nucleotide-free state (Fig. 1a, b), which is consistent with the 49% identity and 77% similarity between the sequences of hABCA4 and hABCA1 (Supplementary Fig. 4).

### Structural features of ABCA4.

The TMDs of ABCA4 exhibited a typical type II ABC exporter fold without TM swapping like other A and G-subfamily ABC transporters[20,40,41]. The discretely folded TMD1 and TMD2, each consisting of 6 TMs, contact with each other in the cytoplasmic membrane leaflet through a narrow interface between TM5 and TM11 (Fig. 2a). However, other than the limited contact areas between TM5 and TM11, all the other side surfaces of TMDs are completely exposed to both leaflets of the lipid bilayer in the current conformation (defined as lateral-opening conformation), a unique structural feature among all ABC transporters and first observed in the nucleotide-free ABCA1 structure (Supplementary Fig. 5a, b).

The cytoplasmic NBDs and RDs, assigned as poly-Ala in the ABCA1 structure (PDB 5XJY), were clearly resolved in our ABCA4 structure, allowing side-chain assignment for most of the NBDs and part of the RDs (Supplementary Figs. 2e and 3a). Despite more structural elements of NBDs and RDs were resolved in ABCA4, the core architecture of these domains is quite similar between ABCA4 and ABCA1 (Supplementary Fig. 5a), consistent with NBDs and RDs as the most conserved elements among ABCA subfamily[6]. The two similar RDs, each comprised of ~80 amino acids folding into three α helices and four anti-parallel

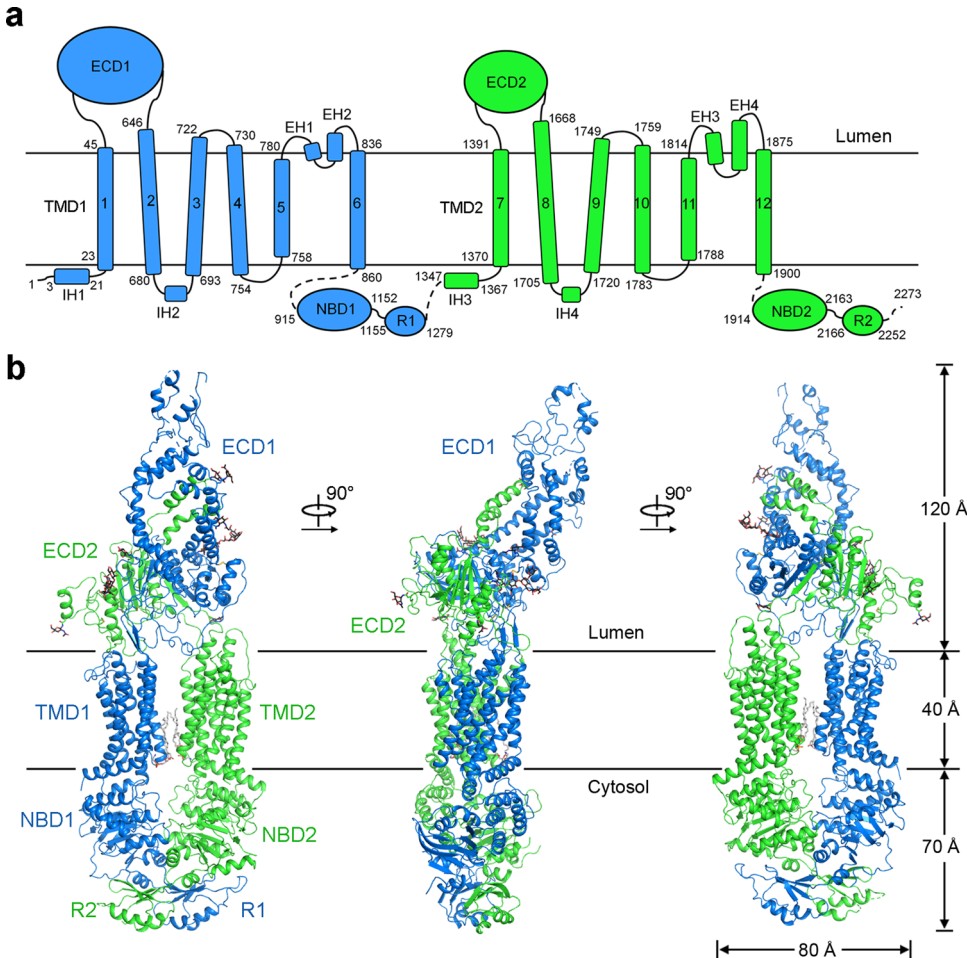

**Fig. 1 Overall structure of human ABCA4. a** Topological diagram of ABCA4. TMD: transmembrane domain; ECD, extracellular domain; NBD, nucleotide-binding domain; R1 and R2, regulatory domains; IH, intracellular helices; EH, extracellular helices. **b** Overall structure of ABCA4. The two halves of ABCA4 are colored marine and green for half 1 (TMD1, ECD1, NBD1 and RD1) and half 2 (TMD2, ECD2, NBD2 and RD2), respectively. Two phospholipid molecules bound between TMD1 and TMD2 in the cytoplasmic membrane leaflet are shown as gray sticks. The N-linked glycans are displayed as black sticks. All structural figures were prepared using PyMol (https://www. pymol.org/) or UCSF Chimera[65].

β-strands, closely contact with each other, which might help to keep the two NBDs close together even before ATP binding (Supplementary Fig. 5c).

The ECD of ABCA4 shares a common architecture as that of ABCA1, which can be divided into three layers, the lid, the tunnel, and the base (Supplementary Fig. 6a). In addition, a similar hydrophobic tunnel was observed within the ECD of ABCA4, which is filled with a number of lipid-like densities that might belong to lipids or detergents (Supplementary Fig. 6b). Despite the similar architecture, the ECDs of ABCA4 and ABCA1 exhibited substantial structural differences (Supplementary Fig. 6c). Except for a roughly 30° upward rotation of the ABCA4 lid compared to that of ABCA1, the major structural difference in the ECDs is the enlarged tunnel in ABCA4, which is contributed by the outward movement of tunnel-forming helices (Supplementary Fig. 6c), suggesting potentially inherent flexibility of the tunnels. Another evident difference between both ECDs is the observation of an extended loop (named S-loop for substrate binding that will be discussed in the NRPE-binding site) in ECD1 of ABCA4, which was not resolved in the ABCA1 structure (Supplementary Fig. 6d). Nevertheless, the structures of the base region are largely identical between ABCA4 and ABCA1 with a root-mean-square-deviation (RMSD) of 1.04 Å over 272 aligned Cα atoms (Extended Date Fig. 6e).

**Two phospholipids bound to the TMDs in the cytoplasmic leaflet.** During structural refinement, two clear phospholipid densities were observed in the cytoplasmic leaflet within the TMDs (Fig. 2b and Supplementary Fig. 3b). The two phospholipids (PL1 and PL2) were found in the shallow pockets mainly enclosed by the cytoplasmic segments of TMs 1/2/5 and TMs 7/8/11, respectively (Fig. 2a). Since the presence of PE and PC molecules in our protein samples (Supplementary Fig. 1c, d) and the phospholipid head groups cannot be distinguished at current resolution, we tentatively modeled these densities as PE in our structures.

The polar head of PL1 is coordinated by the side chains of a cluster of polar and charged residues, including Gln21, Arg24, Lys672, and His1017, whereas the acyl chains are coordinated by TMs 1/2/5/11 via extensive hydrophobic interactions (Fig. 2c, left). The polar head of PL2 is coordinated by the side chains of Lys1371 and Gln1703, whereas its acyl chains are coordinated by TMs 5/7/8/11/ via extensive hydrophobic interactions (Fig. 2c, right). To determine the effects of phospholipid binding, we performed mutagenesis studies. The R24A/K672A/H1017A variant, designed to disrupt the PL1 binding, displayed significantly decreased ATPase activity, whereas the K1371A/Q1703A variant, designed to disrupt the PL2 binding, maintained a similar ATPase activity (Fig. 2d and Supplementary Fig. 3c).

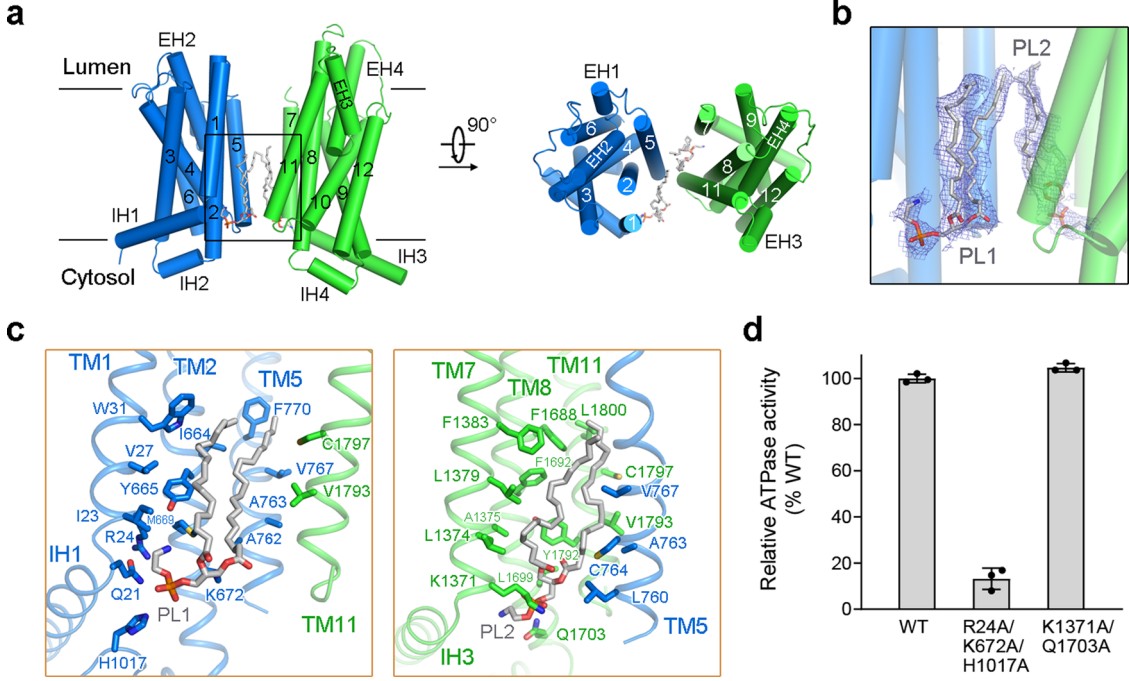

**Fig. 2 Two phospholipids bound to the TMDs in the cytoplasmic leaflet. a** The TMDs of ABCA4 exhibit a lateral-opening conformation with two phospholipids bound in the cytoplasmic leaflet. TMD1 and TMD2 are colored marine and green, respectively. The phospholipids are shown as gray sticks. **b** Electron density maps of the two bound phospholipids. The density maps, shown as blue mesh, are contoured at 3σ. **c** Close-up views of the two phospholipid-binding sites. Key residues that mediate the interactions are shown as sticks. Two clusters of polar and charged residues near the membrane cytoplasmic leaflet might contribute to coordinate the polar heads of phospholipids. **d** Functional verification of key residues that might be involved in coordinating the polar heads of phospholipids. The ATPase activities of two ABCA4 variants were normalized relative to that of the wild-type (WT) ABCA4. For all dot-plot graphs, each data point is the average of three independent experiments, and error bars represent the s.d. Source data for graph (**d**) are provided as a Source Data file.

The results suggest the functional importance of PL1 binding but not PL2 binding.

**NRPE substrate recognition.** ABCA4 has been reported to bind the physiological substrate NRPE with an apparent $K_d$ of 2–5 μM[29]. To reveal the structural basis of substrate recognition by ABCA4, we next determined the structure of WT hABCA4 in the presence of NRPE at an overall resolution of 3.4 Å (Supplementary Fig. 7). Within the TMDs, except the densities for the two phospholipids bound in the cytoplasmic leaflet, two additional lipid-like densities were clearly resolved in the luminal leaflet (Fig. 3a and Supplementary Fig. 8a). The shape of one lipid-like density is consistent with a NRPE, showing one ATR contiguous with two acyl chains, and the other density can be well fitted with a phospholipid molecule (PL3, modeled as PE) (Supplementary Fig. 8b). The overall structure of the NRPE-bound ABCA4 is essentially identical to the apo ABCA4 structure, except for an obvious movement of the loop connecting TM7 and ECD2 toward the PL3 in the NRPE-bound structure (Supplementary Fig. 8c).

The NRPE and PL3 molecules are sandwiched between the two TMDs in the luminal leaflet, but completely exposed to the lipid bilayer, suggesting the lateral access of the lipid substrates from the luminal membrane leaflet (Fig. 3b). The phosphate group of NRPE is mainly recognized by the side chains of two positively charged Arg residues, Arg587 in the β6–α19 loop of ECD1 and Arg653 in TM2, whereas the acyl chains are coordinated by TMs 1/2/5 via extensive hydrophobic interactions (Fig. 3c, left). The ATR moiety of NRPE is coordinated by four aromatic residues (Trp339, Tyr340, Tyr345, and Phe348) in an extended loop of ECD1 (named S-loop for substrate

binding), and the β-ionone ring of the ATR moiety is further coordinated by several hydrophobic residues in TM8 and TM11 (Fig. 3c, right). The residues for NRPE coordination bear similar configurations in both apo and NRPE-bound ABCA4 structures, and only minor local changes occur to Arg587, Arg653, Tyr345, and Phe348 that may be caused by NRPE binding (Supplementary Fig. 8d). Consistent with the structural observations, a recent biochemical study suggested that Arg653 can contribute to NRPE binding[42].

ATR has been shown to stimulate the ATPase activity of ABCA4 in the presence of PE but not PC, suggesting the coupling of NRPE binding to ATP hydrolysis[39,42–44]. We also observed the ATR-stimulated ATPase activity of ABCA4 when the protein was purified in CHAPS/lipid mixture or reconstituted in liposome or nanodisc (Supplementary Fig. 9a–c). The basal ATPase activity of WT ABCA4 in CHAPS/lipid mixture was stimulated up to ~2-fold by ATR, with half-maximal stimulation at ~10 μM ATR (Supplementary Fig. 9a). The difference of ATPase activity in liposome between the current study and the previous ones[39,43,45] might be caused by the different detergents used during protein purification, different lipid compositions and methods of liposome preparation, or the method of protein concentration determination. Similarly, the ATPase activity of GDN purified ABCA4 was stimulated by NRPE (Supplementary Fig. 9d). To reveal the functional importance of NRPE-coordinating residues in ABCA4, we generated nine ABCA4 variants with point mutations designed to disturb NRPE binding, including R587A, R653C, and R587A/R653C designed to disrupt the polar interactions with the phosphate group of NRPE; W339E/Y340E, W339A/Y340A, Y345E/F348E, and Y345A/F348A designed to disrupt the hydrophobic interactions with the ATR moiety

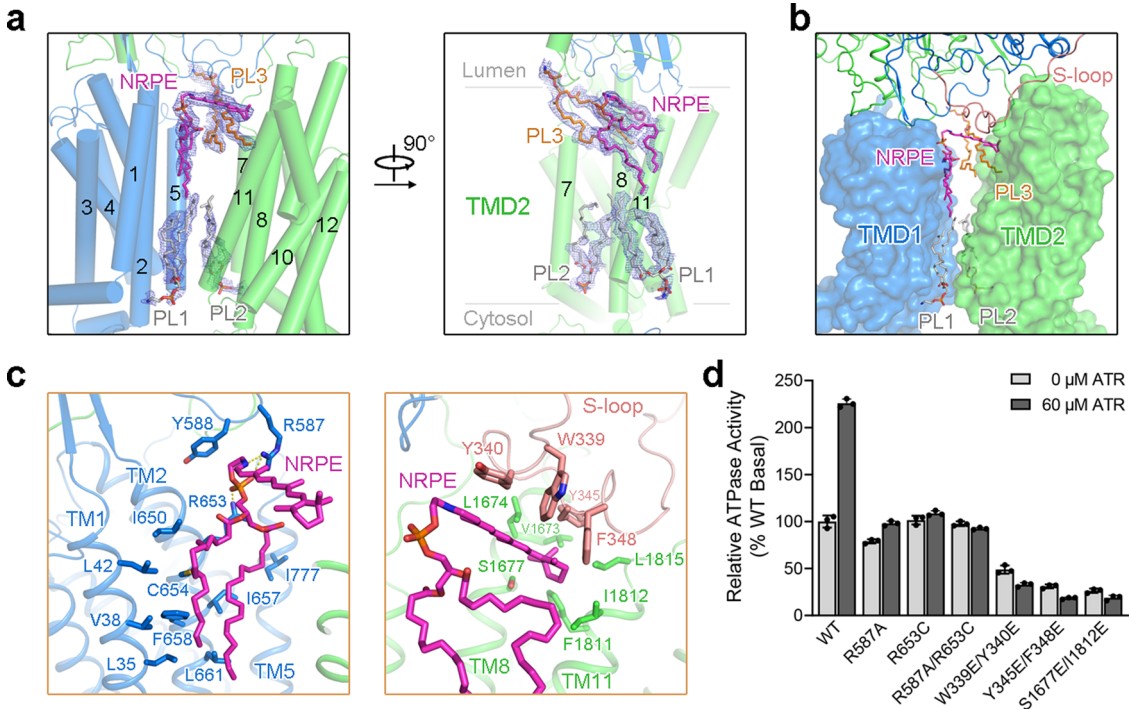

**Fig. 3 NRPE substrate binding site. a** Electron density maps of NRPE and phospholipids in the TMDs of NRPE-bound ABCA4 structure. The density maps, shown as blue mesh, are contoured at 4σ. TMD1 and TMD2 are colored marine and green, respectively. The phospholipids in the cytoplasmic leaflet are shown as gray sticks. The NRPE and phospholipid in the luminal leaflet are colored magenta and orange, respectively. **b** Exposure of NRPE and phospholipids to the lipid bilayer. The S-loop from ECD1, in close proximity to the transmembrane region and involved in NRPE coordination, is highlighted in light pink. **c** Close-up views of the NRPE-binding site. Residues mediating the interactions are shown as sticks. Potential polar interactions are represented by yellow dashed lines. **d** Functional verification of key residues involved in NRPE coordination evaluated by the ATR-stimulated ATPase activity of ABCA4. The ATPase activity was normalized relative to the basal ATPase activity of the WT ABCA4. Each data point is the average of three independent experiments, and error bars represent the s.d. Source data for graph **d** are provided as a Source Data file.

of NRPE; and S1677E/I1812E and S1677A/I1812A designed to explore the interactions with the β-ionone ring of the ATR moiety (Fig. 3d and Supplementary Fig. 9e, f). Except for the S1677A/I1812A variant, which has little influence on the interactions with NRPE, the other eight variants displayed nearly abolished ATR-stimulated ATPase activity (Fig. 3d and Supplementary Fig. 9f), consistent with the involvement of these residues in NRPE coordination.

**Structure of the ATP-bound ABCA4**. To visualize ATP-driven conformational rearrangements of ABCA4, we characterized the structure of ABCA4$_{QQ}$, a mutation that abolished ATP hydrolysis but maintained ATP binding, in the presence of 10 mM ATP/Mg$^{2+}$ (Supplementary Fig. 10). The structure was resolved at an overall resolution of 3.3 Å, showing excellent side-chain density for most of the protein except the lid and RD regions (Supplementary Fig. 10e). The overall structure of the ATP-bound ABCA4 is substantially different from the nucleotide-free ABCA4 structures (Fig. 4a). Two strong and well-defined EM densities were observed for two bound ATP-Mg$^{2+}$ molecules that were sandwiched between the two NBDs (Fig. 4b). The two NBDs exhibit a typical "head-to-tail" configuration predicted for all ABC transporters, forming two ATP binding-sites between the Walker A motif of one NBD and the ABC signature motif of the other NBD (Fig. 4c). In both ATP-binding sites, the phosphate groups of ATP form extensive polar interactions with the Walker A/B motifs, the Q-loop and the H-loop of one NBD, and the ABC signature sequence of the other NBD, and the adenine moiety of ATP forms a π-stacking interaction with the side chain of a conserved aromatic residue in the A-loop (Fig. 4d).

**Conformational changes of ABCA4 upon ATP binding**. The V-shaped lateral-opening conformation of the TMDs in the nucleotide-free ABCA4 structures is completely altered upon NBD dimerization induced by ATP binding (Fig. 5a). In the ATP-bound state, the NRPE and phospholipid-binding cavities in both membrane leaflets have been completely collapsed, with no space for bound molecules (Fig. 5a). The two TMDs in the ATP-bound state form a closed conformation owing to the extensive interactions between TMs 1/2/5 and TMs 7/8/11, in contrast to a narrow interface in the cytoplasmic membrane leaflet between TM5 and TM11 in the nucleotide-free state (Fig. 5a). The transition between the nucleotide-free and ATP-bound conformations can be described as roughly rigid-body rotations of the two TMD-NBD-RD modules, TMD1-NBD1-RD1 and TMD2-NBD2-RD2 (Fig. 5b, c). The two TMD-NBD-RD modules are largely unchanged in the two different states with only some evident local structural rotations for the RecA-like domain of NBD1 and the two RDs (Fig. 5b, c). Due to ATP binding, both modules have rotated toward the molecular center to form the closed TMDs and also the more compact NBDs and RDs (Fig. 5a).

Since the ECD sit just above the luminal face of the TMDs, the ATP-induced conformational changes of TMDs were further translated to the ECD (Fig. 5d). As the electron density for the lid of ECD is relatively poor (Supplementary Fig. 11a, b), the model for lid might not be accurate. Therefore, all the following analyses about the conformational changes regarding ECD do not include the lid region. The co-folded ECD moves as one conjoined rigid body from the nucleotide-free state to the ATP-bound state since the overall structure of the entire ECD is nearly identical in both states, with RMSD of 0.587 Å over 600 Cα atoms (Fig. 5e).

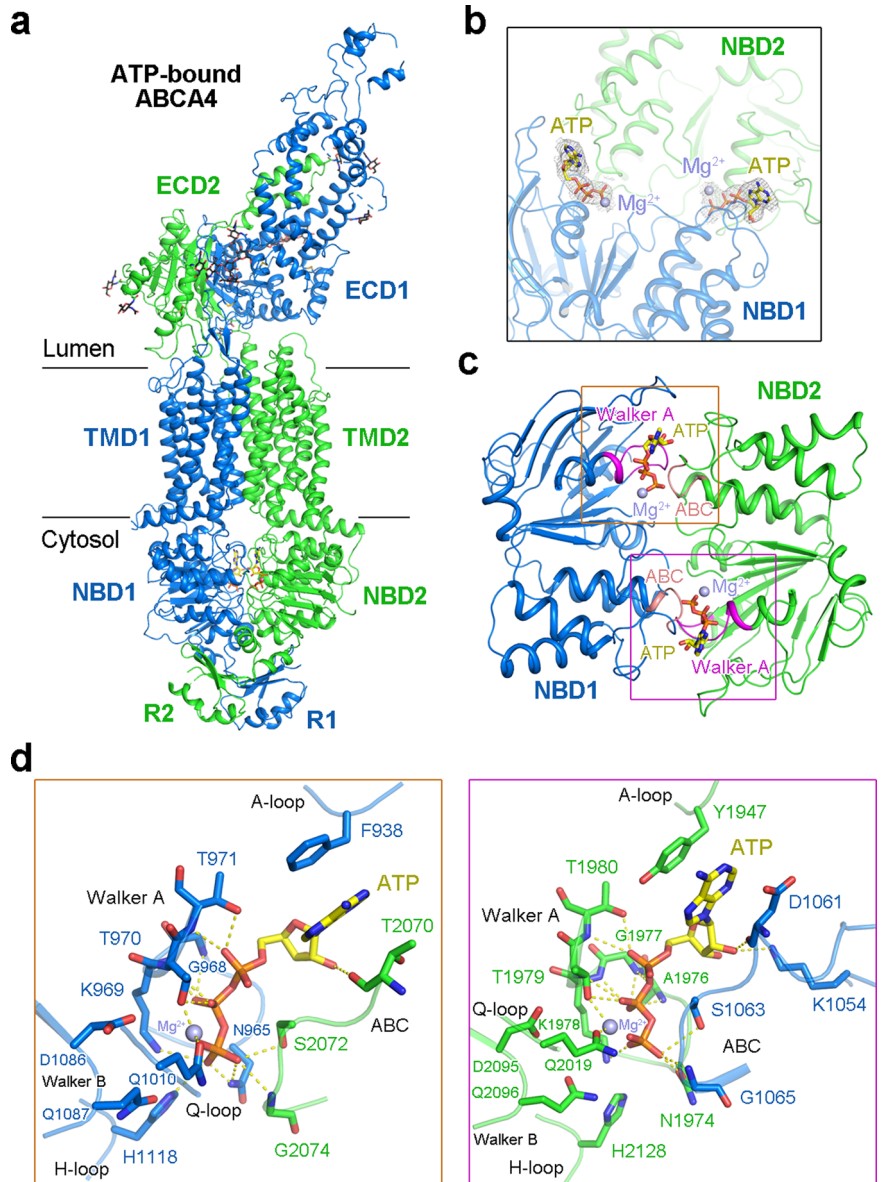

**Fig. 4 Structure of ATP-bound ABCA4. a** Overall structure of ATP-bound ABCA4. The domains are colored as above. **b** Electron density maps of ATP-$Mg^{2+}$ bound between the NBDs. The density maps, shown as gray mesh, are contoured at 5σ. ATP molecules are shown as yellow sticks; $Mg^{2+}$ ions are displayed as purple spheres. **c** The NBD motifs involved in ATP-$Mg^{2+}$ coordination. The Walker A motifs and ABC signature sequences are highlighted in magenta and light pink, respectively. **d** Close-up views of the two ATP-binding sites. Key residues mediating the interactions are shown as sticks. Potential polar interactions are represented by yellow dashed lines.

The S-loop, which participates in NRPE substrate recognition in the nucleotide-free state, was not resolved in the ATP-bound state (Fig. 5e), suggesting the intrinsic flexibility of the S-loop during the transport cycle. After ATP binding, the ECD exhibits an obvious rotation toward the membrane, with a displacement for ~20 Å observed at the far end of the tunnel region (Fig. 5d).

## Discussion

The structural analyses of ABCA4 in three different functional states presented here suggest a model of NRPE translocation by ABCA4 (Fig. 6). In the nucleotide-free and substrate-free state, the TMDs exhibit a V-shaped lateral-opening conformation, allowing the lateral entry of NRPE from the luminal membrane leaflet (state 1). NRPE is recognized through two positively charged Arg residues in TMD1 and ECD1 (Arg653 and Arg587), and coordinated by two TMDs and S-loop of ECD1 through

extensive hydrophobic interactions (state 2). Upon binding of ATP, the rearrangement of TMDs accompanied with ATP-induced NBD closure transmits the TMDs to a closed conformation, which precludes the NRPE binding (state 3). Considering the physiological function of ABCA4 is to decrease the accumulation of NRPE in the luminal leaflet of the disc membranes, we suggest that the ATP-bound ABCA4 would extrude the NRPE from the luminal leaflet to the cytoplasmic leaflet of the membrane, which would be followed by NRPE hydrolysis and RDH-mediated ATR reduction to ATRol (Fig. 6). Our study also suggests that ATP binding might be sufficient for the NRPE-extrusion step and that ATP hydrolysis might be required to reset the transporter to the lateral-opening conformation ready for the next transport cycle (Fig. 6). The NRPE translocation pathway and how NRPE is flipped from the luminal leaflet to the cytoplasmic leaflet remain to be explored in the future.

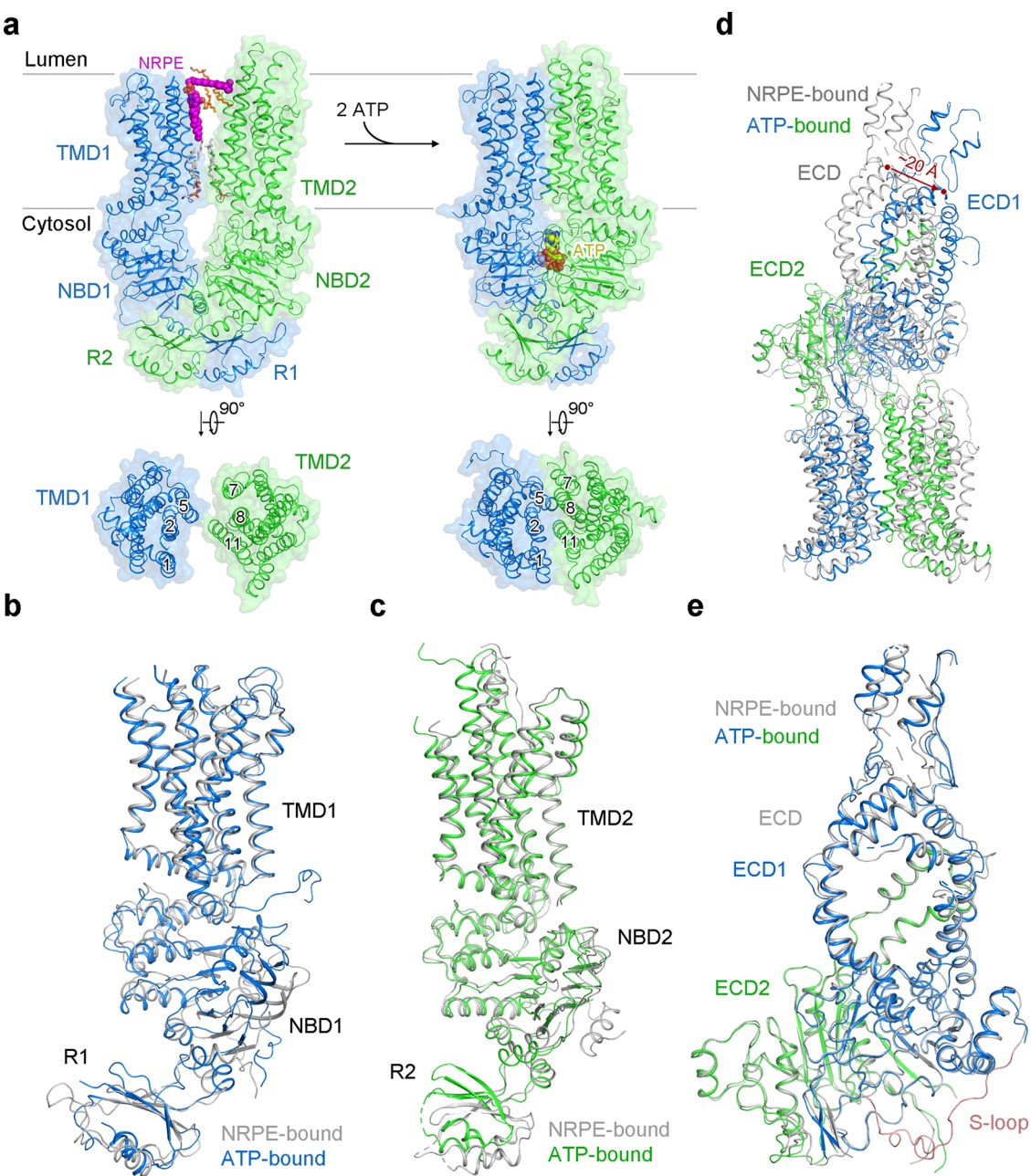

**Fig. 5 Conformational changes of ABCA4 upon ATP binding. a** Closure of the ABCA4 TMDs upon ATP binding. The phospholipids are shown as sticks. NRPE and ATP are displayed as magenta and yellow spheres, respectively. $Mg^{2+}$ ions are shown as purple spheres. **b** Superposition of the TMD1-NBD1-R1 domains of the NRPE-bound ABCA4 (gray) and ATP-bound ABCA4 (marine) structures. **c** Superposition of the TMD2-NBD2-R2 domains of the NRPE-bound ABCA4 (gray) and ATP-bound ABCA4 (green) structures. **d** Rigid-body movement of the ECD upon ATP binding. The NRPE-bound ABCA4 structure was superimposed with the ATP-bound ABCA4 structure. The NBDs and RDs were omitted for clarity. NRPE-bound ABCA4 structure is colored gray. ATP-bound ABCA4 structure is colored based on the domains, with TMD1-ECD1 and TMD2-ECD2 colored marine and green, respectively. The ECD shows an obvious rotation toward the membrane with a displacement for ~20 Å observed at the far end of the tunnel region. **e** Superposition of the ECDs of the NRPE-bound ABCA4 (gray) and ATP-bound ABCA4 (marine and green) structures. The S-loop from NRPE-bound ABCA4 structure is colored light pink. The corresponding region from ATP-bound ABCA4 structure was missing due to flexibility.

During the preparation of this manuscript, two similar ABCA4 structures resolved in apo and ATP-bound states were published[46]. Except for the flexible lid region, both structures are almost identical to the corresponding structures presented by us (Supplementary Fig. 11c). Although these cryo-EM structures provide a vast amount of information about the conformational transitions of ABCA4 to flip NRPE, the structures also raised several unanswered questions as topics for future investigation. For example, what's the role of the ECD for ABCA4 function? A large number of disease-associated mutations in ECD indicate a vital functional role of ECD[38,44,47,48]. It has been suggested that the hydrophobic tunnel in the ECD of ABCA1 might serve as a potentially temporary storage or delivery passage for lipid substrates[20,49]. Considering the highly conserved sequences and structures between ABCA1 and ABCA4, it's logical to infer that the function of the ECD might be conserved between the two proteins. Consist with this, it has been reported that the ECD2 of ABCA4 can specifically interact with ATR[50]. Nevertheless, the

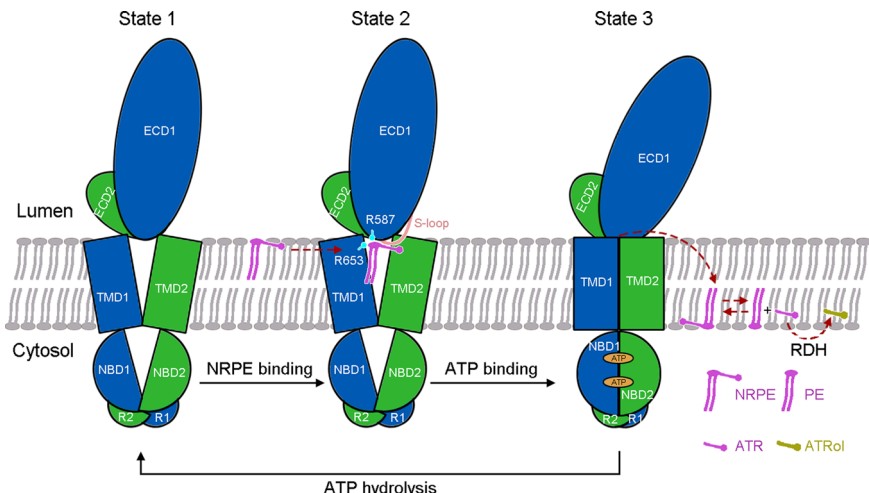

**Fig. 6 A working model of ABCA4-mediated NRPE translocation.** Schematic illustration of the 'lateral access and extrusion' cycle of ABCA4 inferred from the three resolved structures. The two halves of ABCA4 are colored marine and green for half 1 and half 2, respectively. In the absence of ATP, the TMDs exhibit a V-shaped lateral-opening conformation, allowing the lateral access of the NRPE substrate from the luminal membrane leaflet. ATP binding transmits the TMDs to a closed conformation, which might extrude the NRPE from luminal leaflet to cytoplasmic leaflet followed by NRPE hydrolysis and retinal dehydrogenase (RDH)-mediated ATR reduction to ATRol. ATP hydrolysis resets the transporter to the lateral-opening conformation ready for next transport cycle. See text for discussion of the proposed working model.

precise functional role of the ABCA4-ECD requires careful examination in the future.

Another intriguing question raised by our structural study is the functional role of the phospholipids bound in the TMDs. Further studies would be required to reveal whether there is phospholipid specificity for these binding sites and whether the phospholipids would be translocated across the membrane during the NRPE transport cycle.

In the ABCA4-mediated NRPE translocation model, ABCA4 functions as an importer translocating the substrate from the luminal leaflet to the cytoplasmic leaflet of membrane (Fig. 6). This transport direction, which has been directly demonstrated in a biochemical reconstitution system[30], is distinct from almost all the other mammalian ABC transporters thought to function in the export direction[4,51]. Especially, the other ABCA transporters, such as ABCA1 and ABCA7, have been proved to export the phospholipids across the membranes[38].

The TMDs of ABCA1 and ABCA4 exhibit a conserved lateral-opening conformation exposing to both membrane leaflets in the nucleotide-free state, which would allow the lateral access of the substrates from both membrane leaflets (Supplementary Fig. 5b). This unique conformation of the TMDs might explain the seemly controversial transport direction mediated by a family of conserved transporters, since the lipid substrates might gain access to the substrate binding cavities in TMDs from different membrane leaflets for different ABCA transporters. The similarity of their substrates and also the common architecture of ABCA transporters indicate that they might adopt a conserved 'lateral access and extrusion' mechanism to transport the lipid substrates across the membrane. The substrate transport directionality might be determined by the asymmetric distribution of substrates on both membrane leaflets and the different substrate binding affinities with the TMD in the inner and outer membrane leaflets. Further structural studies of different ABCA transporters in complex with their lipid substrates and in ATP-bound states are necessitated to establish whether a universal mechanism exists for ABCA transporters.

## Methods

**Protein expression and purification.** The WT and variants of human ABCA4 were cloned into the pCAG vector either with an N-terminal Flag-tag and a C-

terminal His-tag or with a C-terminal Flag-His tandem tag. A complete list of all primers used in this study has been supplied in Supplementary Table 2. The HEK293F suspension cells (Invitrogen) were cultured in SMM 293T-II medium (Sino Biological Inc.) at 37 °C under 5% CO2 in a shaker. The cells were transiently transfected with the expression plasmid and polyethyleneimine (PEI) (Polysciences) at a density of $2.0–2.5 \times 10^6$ cells per ml. About 1 mg of plasmid was incubated with 3 mg of PEI for 15–30 min at room temperature before being applied to each liter of cell culture to initiate transfection. After infection for 12 hr, 10 mM sodium butyrate was supplemented to boost protein expression. The cells were further cultured for 48 hr before being collected.

The cell pellet was resuspended in buffer containing 50 mM Tris pH 7.5, 150 mM NaCl and protease inhibitor cocktail (Amresco). The cell membrane was solubilized with 1% (w/v) DDM and 0.2% (w/v) cholesteryl hemisuccinate (CHS) at 4 °C for 2 hr. After centrifugation at $37,000 \times g$ for 1 h, the supernatant was collected and applied to anti-Flag G1 affinity resin (GenScript). The resin was rinsed with wash buffer (W1 buffer) containing 50 mM Tris pH 7.5, 150 mM NaCl, and 0.02% GDN. The protein was eluted with W1 buffer supplemented with 200 μg/ml Flag peptide. The eluent was further applied to the Ni-NTA resin (QIAGEN), which was subsequently washed with W2 buffer (50 mM Tris pH 7.5, 150 mM NaCl, 20 mM imidazole and 0.02% GDN) and eluted with W2 buffer supplemented with 230 mM imidazole. The protein eluent was concentrated and further purified by size-exclusion chromatography (SEC) using a Superose 6 Increase 10/300 GL column (GE Healthcare) equilibrated with W1 buffer. Peak fractions were pooled and concentrated for biochemical studies or cryo-EM experiments.

A similar protein purification strategy was applied to prepare the ABCA4 protein in CHAPS/lipid mixture for measuring the ATR-stimulated ATPase activity. The cell pellet was resuspended in buffer containing 50 mM Tris pH 7.5, 150 mM NaCl, 6 mM $MgCl_2$, 10% glycerol, and protease inhibitor cocktail. The cell membrane was solubilized with 20 mM CHAPS in the presence of 1 mM DTT at 4 °C for 2 h. After centrifugation, the supernatant was collected and applied to anti-Flag affinity chromatography, followed by nickel affinity chromatography. The protein was eluted in the buffer R1 containing 50 mM Tris pH 7.5, 150 mM NaCl, 10 mM CHAPS, 6 mM $MgCl_2$, 1 mM DTT, 10% glycerol, 0.19 mg/ml brain polar lipid (BPL), 0.033 mg/ml DOPE, and 250 mM imidazole. The protein eluent was ready for malachite green ATPase assay which will be introduced below.

**Proteoliposome and nanodisc reconstitutions.** The ABCA4 protein purified in W1 buffer was used for proteoliposome and nanodisc reconstitutions. Equal amounts of BPL and DOPE, both dissolved in chloroform, were mixed and dried under a nitrogen stream. The mixture was resuspended in W1 buffer to a final concentration of 5 mM. After sonication for 30 min and 10 freeze-thaw cycles, the BPL/DOPE (1:1, w/w) solution should be ready for use. Proteoliposome was prepared by mixing ABCA4 and BPL/DOPE (1:1, w/w) with a molar ratio of 1:3000. The detergent was removed by manually packed Sephadex G-50 (Sigma G5050) column pre-equilibrated with buffer R2 (50 mM Tris pH 7.5 and 150 mM NaCl). The proteoliposome fractions were manually collected for biochemical studies. Nanodisc was prepared by mixing ABCA4, MSP1E3D1, and BPL/DOPE (1:1, w/w) with a molar ratio of 1:10:1000, followed by detergent removal with Bio-beads

(Bio-Rad). The reconstituted ABCA4 nanodisc was further purified by a Superose 6 Increase 10/300 GL column (GE Healthcare) equilibrated with buffer R2. The peak fractions were collected and concentrated for biochemical analyses.

**ATPase activity assay**. The ATPase activity of ABCA4 in GDN micelles was measured using a continuous spectrophotometric assay based on the coupling of ADP production and NADH oxidation by pyruvate kinase (PK) and lactate dehydrogenase (LDH)[52]. The assay was carried out at 37 °C on a 150-μL scale in 96-well micro-plates. The reaction mixture contained 50 mM Tris pH 7.5, 150 mM NaCl, 0.02% GDN, 60 μg/ml PK, 32 μg/ml LDH, 3 mM phosphoenolpyruvate (PEP), 0.4 mM NADH, 2 mM DTT, 0.1 μM ABCA4 and varying amounts of ATP/MgCl$_2$. Measurements were taken at 340 nm based on the decrease of NADH absorbance over the course of one hour on SYNERGY H1 microplate reader (BioTek). Statistical analysis was performed using GraphPad Prism 8. The initial rate of ATP hydrolysis versus ATP concentration was fitted with the Michaelis-Menten equation.

As ATR interferes with the NADH-coupled ATPase assay, the malachite green ATPase assay was chosen as an alternative to determine the ABCA4 ATPase activity in the presence of ATR performed similarly as previously described[53]. The assay was performed on a 180-μL scale and shielded from light. To generate NRPE from PE and ATR, 0.1 μM ABCA4 purified in buffer R1 (CHAPS/lipid mixture) or reconstituted in liposome or nanodisc was pre-incubated with ATR at RT for 30 min. The mixture was then supplemented with 2 mM ATP, and the reaction was carried out at 37 °C for 30 min. The reaction mixture was diluted 4 times with water before being mixed with fresh developing solution (2.4 M H$_2$SO$_4$, 1 mg/ml malachite green, 1.5% ammonium molybdate, and 0.2% Tween 20) at a volume ratio of 50:12. The mixture was incubated at RT for 30 min before being detected in 96-well micro-plates at 623 nm on SYNERGY H1 microplate reader. Statistical analysis was performed using GraphPad Prism 8.

**Thin-layer chromatography (TLC)**. The TLC method was used to characterize the endogenous phospholipids bound to the purified ABCA4 protein. Chloroform and methanol were mixed with 0.15 mg ABCA4 protein with a volume ratio of 1:2:2 for lipid extraction. The lower chloroform phase was collected and dried under a nitrogen stream. The lipids were re-dissolved in 25 μl of chloroform and 8 μl sample was applied onto a Silica TLC plate. The lipids were separated by a solvent system containing chloroform, methanol and ammonium hydroxide (65:25:4, v/v/v). The TLC plate was dried and stained with iodine vapor for visualization of the lipids.

**Mass spectrometry analysis**. LC-MS/MS was applied to further identify the endogenous phospholipids bound by purified ABCA4 protein. The lipids were extracted similarly as above and resuspended in the ACN/IPA/H$_2$O (65:30:5, v/v/v) solution containing 5 mM ammonium acetate. The sample was injected into the LC-MS/MS system (Q Exactive Orbitrap Mass Spectrometer, Thermo Scientific) equipped with a C18 reversed-phase column (1.9 μm, 2.1 mm × 100 mm, Thermo Scientific). Mobile phases A and B were ACN/H$_2$O (60:40, v/v) and IPA/ACN (90:10, v/v), respectively, both supplemented with 10 mM ammonium acetate. The elution gradient was started with 32% B for 1.5 min, and then linearly increased to 85% B at 15.5 min. The gradient was raised to 97% B at 15.6 min, and held for 2.4 min, before being dropped back to 32% B at 18.1 min for column equilibration (total program length of 22 min). The mass spectrometer was operated in negative mode with a spray voltage of 3.5 kV. Full scan mass spectrometry data were obtained from the m/z range of 166.7−2000. The MS/MS spectra were searched against the lipid database by LipidSearch software (Thermo Scientific).

**Cryo-EM sample preparation and data collection**. For cryo-EM sample preparation of apo ABCA4, 3.5-μL aliquots of purified WT ABCA4 (~10 mg/ml) were applied to the glow-discharged Quantifoil Cu R1.2/1.3 300 mesh grids. After being blotted for 4.5 s, the grids were flash-frozen in liquid ethane cooled by liquid nitrogen with Vitrobot (Mark IV, Thermo Fisher Scientific). For the NRPE-bound ABCA4 sample, ~10 mg/ml WT ABCA4 was pre-incubated with 1.3 mM NRPE on ice for 1 hr before being applied to the grids. The NRPE was prepared by mixing equal moles of DOPE (Avanti 850725 P; dissolved in chloroform) and ATR (Sigma R2500; dissolved in ethanol), and incubating at RT for 30 min with light shielding. The mixture was dried under a nitrogen stream before being resuspended with 1% sodium cholate to a final concentration of 13.75 mM. After sonication at RT for 30 min, the NRPE stock should be ready for use. For the ATP-bound ABCA4 sample, ~10 mg/ml ABCA4$_{QQ}$ was mixed with 10 mM ATP/MgCl$_2$ on ice for 1 hr before grid preparation.

For the apo ABCA4 sample, a total of 4,331 micrograph stacks were automatically collected with SerialEM[54] on a Titan Krios microscope at 300 kV equipped with a K2 Summit direct electron detector (Gatan) and a GIF Quantum energy filter (Gatan) with a slit width of 20 eV, at a nominal magnification of ×130,000 with defocus values from −2.0 to −1.0 μm. Each stack was exposed in super-resolution mode for 5.76 s with an exposure time of 0.18 s per frame, resulting in 32 frames per stack. The total dose for each stack was 50 e$^-$/Å$^2$. For the NRPE-bound ABCA4 and ATP-bound ABCA4 samples, 3,435 micrograph stacks and 3,572 micrograph stacks were individually collected in the same manner. The

stacks were motion corrected with MotionCor2[55] with a binning factor of 2, resulting in a pixel size of 1.08 Å, meanwhile dose weighting was performed[56]. The defocus values were estimated using Gctf[57].

**Cryo-EM data processing**. For the apo ABCA4 dataset, a total of 2,554,957 particles were automatically picked from 4,331 micrographs using Relion 3.0[58]. After 2D classification, 1,761,864 particles were selected and subjected to global search 3D classification. The map of human ABCA1 (EMD-6724) was used as the 3D reference for initial 3D classifications. A total of 941,972 particles selected from the global search 3D classification were subjected to 3D auto-refinement. After another round of local search 3D classification, a total of 205,597 particles were selected. 3D auto-refinement of the particles with an adapted mask yielded a reconstruction with an overall resolution of 3.3 Å.

The procedures for NRPE-bound ABCA4 and ATP-bound ABCA4 data processing were similar as above. For the NRPE-bound ABCA4 dataset, 2,227,280 particles were automatically picked from 3,435 micrographs. 2D classification resulted in 1,893,841 good particles that were then subjected to two rounds of global search 3D classification and auto-refinement. After further local search 3D classification with an adapted mask, a total of 184,628 particles were selected. 3D auto-refinement of the particles with mask yielded a 3.4 Å reconstruction. For the ATP-bound ABCA4 dataset, a total of 2,169,513 particles were automatically picked from 3,572 micrographs. 2D classification resulted in 1,410,668 good particles that were then subjected to two rounds of global search 3D classification and auto-refinement. After an additional round of local search 3D classification, a total of 173,278 particles were selected. 3D auto-refinement of the particles with an adapted mask yielded a reconstruction at resolution of 3.3 Å.

All 2D classification, 3D classification, and 3D auto-refinement procedures were performed using Relion 3.0. Resolutions were estimated using the gold-standard Fourier shell correlation 0.143 criterion[59] with high-resolution noise substitution[60].

**Model building and refinement**. A homology model of ABCA4 was generated by the SWISS-MODEL server[61], using the EM structure of human ABCA1 (PDB code 5XJY) as the reference. The model was docked into the 3.3 Å map of apo ABCA4 using Chimera. Each residue was manually adjusted with Coot[62]. Structure refinements were carried out by Phenix[63] in real space with secondary structure and geometry restraints. The structural model was validated by Phenix and MolProbity[64]. The ABCA4 structure was used as the reference for model building of NRPE-bound ABCA4 and ATP-bound ABCA4. The model refinement and validation statistics were summarized in Supplementary Table 1.

**Reporting summary**. Further information on research design is available in the Nature Research Reporting Summary linked to this article.

## Data availability
The EM density maps generated in this study have been deposited in the EMDB under accession codes EMD-31000 (apo ABCA4), EMD-31001 (NRPE-bound ABCA4), and EMD-31002 (ATP-bound ABCA4). The atomic coordinates generated in this study have been deposited in the PDB under the accession codes 7E7I (apo ABCA4), 7E7O (NRPE-bound ABCA4), and 7E7Q (ATP-bound ABCA4). Source data are provided with this paper.

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

## Acknowledgements
We thank the Cryo-EM Facility of Southern University of Science and Technology for providing the facility support. This work was supported by the National Natural Science Foundation of China (92057101, X.G.), the Natural Science Foundation of Guangdong Province of China for Distinguished Young Scientists (2019B151502047, X.G.) and the Shenzhen Science and Technology Program (RCYX20200714114522081, X.G.).

## Author contributions
X.G. conceived and supervised the project. X.G., T.X., and Z.Z. designed experiments. T.X., Z.Z., Q.F. and B.D. performed the experiments. All authors contributed to data analysis. T.X. and X.G. wrote the manuscript.

## Competing interests
The authors declare no competing interests.
