## [Peer Review File · Nature Communications]

REVIEWER COMMENTS

Reviewer #1 (Remarks to the Author):

The manuscript by Gong et al presents three cryo-EM structures of detergent purified ABCA4, an apo form, a substrate (NRPE) bound, and ATP bound. These structures represent key states in the conformational cycle of ABCA4 that has implications for other members of the ABCA family as well. The structural work is straightforward, clearly presented, and beautiful. Accompanying this is ATPase activity data and MS analysis to identify endogenous lipids co-purifying with their protein. Considering the physiological relevance of ABCA4 in age related macular degeneration, insights provided into its substrate transport cycle and associated conformational transitions makes this work, in this reviewer's opinion, appropriate for publication in Nature Communications. That said, the analysis of the manuscript could benefit from clarification on a few points:

1. Did the authors attempt liposome / nanodisc reconstitution of ABCA4 and compare their ATPase results, especially for ATR stimulation, with detergent purified samples? If so, what were the results. If not, why? The former would represent a more physiological state, and any difference would need to be analyzed accordingly.
2. The y axes on the Buffer vs A4 MS spectra give a false impression of how enriched A4 containing samples are in ions assigned as PE/PC. Were any phospholipid standards run to quantitate the results?
3. Why were TLC standards limited to POPE/POPC instead of natural PE or PC comprising different fatty acid compositions? Presumably this would match any endogenous lipids in the sample more clearly.
4. For cryo-EM data processing, what 3D reference was used for initial 3D classifications? The authors mention using a homology model based on ABCA1 for model building only.
5. What was the source of the NRPE for cryo-EM sample preparation? Relatedly, authors should provide vendor/source information for key reagents.
6. For point mutations tested, were the mutants biochemically characterized? Can the the authors exclude effects of altered stability/biochemical properties of the mutants on on altered ATPase activity?
7. As presented, it is unclear to this reviewer whether ATR stimulation was measured in the GDN sample or in CHAPS only? As the cryo-EM sample was prepared in GDN, ATR stimulation in GDN should be shown as well.
8. Authors should more clearly highlight local changes in sidechain positions, if any, for NRPE interacting residues in apo and NRPE complex structures.
9. The authors used a large excess of NRPE for cryo-EM studies. This may be standard practice to ensure maximal occupancy for structural studies but, considering a low μM EC50 for ATR, one would assume bound NRPE could be detected by MS or TLC after desalting/SEC of the NRPE incubated complex? Were any such analyses performed?

Additional/Discussion Points

1. Type 1 exporter enthusiasts may find the statement that ABCA family members are more complicated than others problematic in light of properties like domain swapping, and conformational dynamics arising from TM flexibility in the former compared to the largely rigid body motions associated with A family transporter conformations.
2. The structures provide insight into the basis for alternating access to a potential luminal binding site for NRPE but it is not clear to this reviewer how the key step of translocation/flopping to the inner leaflet would happen through occlusion of the luminal side upon ATP binding. As such it is a bit of an overstatement to say the structural basis for translocation is revealed.
3. The authors point to implications for other ABCA family transporters. While this is true for the adoption of a closed, ATP bound state, that is likely a shared feature, the opposite substrate transport directionality remains problematic to rationalize. The discussion of this point could be expanded.

Reviewer #2 (Remarks to the Author):

The authors determined three cryo-EM structures of human ABCA4, a retinal-specific ABCA transporter, in distinct functional states: apo state, N-retinylidene-phosphatidylethanolamine (NRPE)-bound state, and ATP-bound state. They also determined the amino acid residues important for NRPE recognition. This study presented some new findings which may be important for understanding the function of ABCA4, while it is still unknown what the role of the ECD is or how ABCA4 functions as an importer.

Major comments:

1. ATPase activity of ABCA4 was measured by using ABCA4 purified with CHAPS. ATPase activity should be also analyzed by using the sample purified with GDN to show the protein used for structure analysis was functional.
2. Comparison of the structures determined in this study with those determined by Jue Chen's group (eLife 2021) will be beneficial for readers to understand the structures of ABCA4.
3. The authors suggested that the co-folded ECD moves as one conjoined rigid body from the nucleotide-free state to the ATP-bound state since the overall structure of the entire ECD is nearly identical in both states (page 12). It is an interesting observation. The electron density map of the whole ECD should be shown as an extended data.
4. W339E/Y340E and Y345E/F348E mutants were constructed to disrupt the hydrophobic interactions with the ATR moiety of NRPE; and S1677E/I1812E were designed to disturb the interaction with the β -ionone ring of the ATR moiety in Fig. 3d. Replacement of hydrophobic amino acid residues by glutamate may affect the structure itself. Mild substitution with other amino acid residues such as alanine or valine should be performed.

Minor point

The first line of INTRODUCTION said that Human ABC transporter superfamily, consisting of 49 members -----. It should be 48 not 49.

Response to reviewers' comments:

Reviewer #1:

The manuscript by Gong et al presents three cryo-EM structures of detergent purified ABCA4, an apo form, a substrate (NRPE) bound, and ATP bound. These structures represent key states in the conformational cycle of ABCA4 that has implications for other members of the ABCA family as well. The structural work is straightforward, clearly presented, and beautiful. Accompanying this is ATPase activity data and MS analysis to identify endogenous lipids co-purifying with their protein. Considering the physiological relevance of ABCA4 in age related macular degeneration, insights provided into its substrate transport cycle and associated conformational transitions makes this work, in this reviewer's opinion, appropriate for publication in Nature Communications. That said, the analysis of the manuscript could benefit from clarification on a few points:

We thank the reviewer for his/her positive appraisal. Please find our response to your insightful comments and suggestions below.

Major points:

1. Did the authors attempt liposome / nanodisc reconstitution of ABCA4 and compare their ATPase results, especially for ATR stimulation, with detergent purified samples? If so, what were the results. If not, why? The former would represent a more physiological state, and any difference would need to be analyzed accordingly.

We thank this reviewer for the kind suggestion. The basal ATPase activity of liposome/nanodisc reconstituted ABCA4 is relatively low compared to that of detergent purified samples (~10 nmol/min/mg v.s. ~80-100 nmol/min/mg). However, the ATPase activity of liposome/nanodisc reconstituted ABCA4 could also be stimulated by ATR for about 2-fold, similar to that for detergent purified samples (Figure shown below). The results indicated that the ATR-stimulation of detergent purified samples is functionally relevant. Since we couldn't explain the much lower basal ATPase activity in liposome/nanodisc, we didn't discuss this aspect in the manuscript.

2. The y axes on the Buffer vs A4 MS spectra give a false impression of how enriched A4 containing samples are in ions assigned as PE/PC. Were any phospholipid standards run to quantitate the results?

We thank this reviewer for the critical comment. The y axes on the Buffer vs A4 MS spectra were rescaled to exhibit the same range in Extended Data Fig. 1d (upper panel and middle panel) in the revised manuscript. Additionally, MS spectra data for the phospholipid standards were presented in the lower panel of Extended Data Fig. 1d in the revised manuscript.

3. Why were TLC standards limited to POPE/POPC instead of natural PE or PC comprising different fatty acid compositions? Presumably this would match any endogenous lipids in the sample more clearly.

We thank this reviewer for the kind suggestion. As TLC gives poor resolution for components having similar properties, natural PE or PC with different acyl chain compositions could not be separated by the TLC plate (Figure shown below, BPL represents the brain polar lipid extract). Therefore, MS spectra was further applied to characterize the exact phospholipid species in the manuscript.

4. For cryo-EM data processing, what 3D reference was used for initial 3D classifications? The authors mention using a homology model based on ABCA1 for model building only.

Point taken. Per this reviewer's suggestion, an additional statement "The map of human ABCA1 (EMD-6724) was used as the 3D reference for initial 3D classifications." was added in the revised manuscript (Page 33, line 7-8).

5. What was the source of the NRPE for cryo-EM sample preparation? Relatedly, authors should provide vendor/source information for key reagents.

Point taken. The details for NRPE preparation and related vendor information were added in the revised manuscript (Page 32, line 6-11).

6. For point mutations tested, were the mutants biochemically characterized? Can the authors exclude effects of altered stability/biochemical properties of the mutants on altered ATPase activity?

Point taken. All the mutants were biochemically characterized and displayed similar behavior as the WT protein as shown by the SEC results (Extended Data Fig. 3c and Extended Data Fig. 9d) in the revised manuscript.

7. As presented, it is unclear to this reviewer whether ATR stimulation was measured in the GDN sample or in CHAPS only? As the cryo-EM sample was prepared in GDN, ATR stimulation in GDN should be shown as well.

We are sorry for the confusion. ATR stimulation was only measured in CHAPS purified samples in the original manuscript. Per this reviewer's request, we've determined the ATR stimulation of GDN purified protein sample in the revised manuscript. Since no extra lipid was supplemented for the GDN purified protein, NRPE instead of ATR was used in the ATPase activity stimulation assay. The ATPase activity of GDN purified ABCA4 was stimulated for about 30% by NRPE (Extended Data Fig. 9c). Since the ATR stimulation of CHAPS purified protein is more robust and has been well characterized previously (*Garces et al, 2021, Int J Mol Sci 22:185*), we've used the same assay system to determine the ATR-stimulated ATPase activity of related ABCA4 variants.

8. Authors should more clearly highlight local changes in sidechain positions, if any, for NRPE interacting residues in apo and NRPE complex structures.

We thank this reviewer for the kind suggestion. The local changes in sidechain positions for NRPE interacting residues in apo and NRPE-bound ABCA4 structures were highlighted in Extended Data Fig. 8d in the revised manuscript. A corresponding comment "*The residues for NRPE coordination bear similar configurations in both apo and NRPE-bound ABCA4 structures, and only minor local changes occur to Arg587, Arg653, Tyr345, and Phe348 that may be caused by NRPE binding.*" was added in the revised manuscript (Page 10, line 10-13).

9. The authors used a large excess of NRPE for cryo-EM studies. This may be standard practice to ensure maximal occupancy for structural studies but, considering a low μM EC50 for ATR, one would assume bound NRPE could be detected by MS or TLC after desalting/SEC of the NRPE incubated complex? Were any such analyses performed?

We thank this reviewer for the critical comment. Since the binding characteristics between NRPE and ABCA4 have been thoroughly investigated in *Beharry et al, 2004 (JBC 279:53972)* and *Garces et al, 2021 (Int J Mol Sci 22:185)*, we didn't perform such analyses regarding the interaction in our manuscript. In the previous studies, retinoid was added to the ABCA4-associated resin, and the specifically bounded retinoid was further extracted for detection. Both HPLC and radiolabeling methods have been applied to measure the binding affinity between NRPE and ABCA4. Therefore, a citation of the interaction between ABCA4 and NRPE has been added in the revised manuscript (Page 9, line 10-11).

Additional/Discussion points:

1. Type 1 exporter enthusiasts may find the statement that ABCA family members are more complicated than others problematic in light of properties like domain swapping, and conformational dynamics arising from TM flexibility in the former compared to the largely rigid body motions associated with A family transporter conformations.

Point taken. Per this reviewer's suggestion, the statement "*making them more complicated than the other ABC transporters*" (Page 3, line 10-11 in the original manuscript) was deleted in the revised manuscript.

2. The structures provide insight into the basis for alternating access to a potential luminal binding site for NRPE but it is not clear to this reviewer how the key step of translocation/flopping to the inner leaflet would happen through occlusion of the luminal side upon ATP binding. As such it is a bit of an overstatement to say the structural basis for translocation is revealed.

We thank this reviewer for the critical comment. The structural information presented in our manuscript reveals three key states in ABCA4-mediated substrate translocation, including the pre-substrate-binding state, the substrate-bounded pre-translocation state, and the post-translocation state. As this reviewer mentioned, how the key step of NRPE translocation/flopping to the inner leaflet would happen through occlusion of the luminal side upon ATP binding requires further investigation. To make this clear, we have added a statement "*The NRPE translocation pathway and how NRPE is flipped from the luminal leaflet to the cytoplasmic leaflet remain to be explored in the future.*" in the revised manuscript (Page 14, line 3-5).

3. The authors point to implications for other ABCA family transporters. While this is true for the adoption of a closed, ATP bound state, that is likely a shared feature, the opposite substrate transport directionality remains problematic to rationalize. The discussion of this point could be expanded.

We thank this reviewer for the kind suggestion. Per this reviewer's suggestion, we've expanded the discussion section with "*The substrate transport directionality might be determined by the asymmetric distribution of substrates on both membrane leaflets and the different substrate binding affinities with the TMD in the inner and outer membrane leaflets.*" in the revised manuscript (Page 16, line 1-4).

Reviewer #2:

The authors determined three cryo-EM structures of human ABCA4, a retinal-specific ABCA transporter, in distinct functional states: apo state, N-retinylidene-phosphatidylethanolamine (NRPE)-bound state, and ATP-bound state. They also determined the amino acid residues important for NRPE recognition. This study presented some new findings which may important for understanding the function of ABCA4, while it is still unknown what the role of the ECD is or how ABCA4 functions as an importer.

We thank the reviewer for his/her positive overall evaluation. Please find our response to your constructive suggestions below.

Major comments:

1. ATPase activity of ABCA4 was measured by using ABCA4 purified with CHAPS. ATPase activity should be also analyzed by using the sample purified with GDN to show the protein used for structure analysis was functional.

We are sorry for the confusion. ATPase activity of ABCA4 was measured by using ABCA4 purified with either CHAPS (Extended Data Fig. 9a, b, e) or GDN (Extended Data Fig. 1b), which has been clearly described in the corresponding figure legends in the revised manuscript. The protein purified with GDN used for structural analysis was functional (Extended Data Fig. 1b).

2. Comparison of the structures determined in this study with those determined by Jue Chen's group (eLife 2021) will be beneficial for readers to understand the structures of ABCA4.

Point taken. Apart from the flexible lid region, both the ATP-free and ATP-bound ABCA4 structures presented by Jue Chen's group (eLife 2021) are almost identical to the corresponding structures determined by our group. The structure comparison has been added in the revised manuscript as Extended Data Fig. 11b. A corresponding comment "*Except for the flexible lid region, both structures are almost identical to the corresponding structures presented by us.*" was added in the revised manuscript (Page 14, line 8-10).

3. The authors suggested that the co-folded ECD moves as one conjoined rigid body from the nucleotide-free state to the ATP-bound state since the overall structure of the entire ECD is nearly identical in both states (page 12). It is an interesting observation. The electron density map of the whole ECD should be shown as an extended data.

We thank this reviewer for the kind suggestion. Electron density maps of the whole ECD from apo, NRPE-bound, and ATP-bound ABCA4 structures were presented in Extended Data Fig. 11a in the revised manuscript.

4. W339E/Y340E and Y345E/F348E mutants were constructed to disrupt the hydrophobic interactions with the ATR moiety of NRPE; and S1677E/I1812E were designed to disturb the interaction with the β -ionone ring of the ATR moiety in Fig. 3d. Replacement of hydrophobic amino acid residues by glutamate may affect the structure itself. Mild substitution with other amino acid residues such as alanine or valine should be performed.

We thank this reviewer for the critical comment. The residues mentioned above were replaced with alanine to generate the W339A/Y340A, Y345A/F348A, and S1677A/I1812A mutants. The ATR-stimulated ATPase activity for the three mutants were determined in the revised manuscript (Extended Data Fig. 9e). The ATR stimulation of W339A/Y340A and Y345A/F348A variants were also abolished as the W339E/Y340E and Y345E/F348E variants. The S1677A/I1812A variant maintained the ATR-stimulated ATPase activity as the WT protein. The results suggested that mild substitution of Ser1677 and Ile1812 with alanine has little influence on the shape of the substrate-binding pocket and the related hydrophobic interactions, while replacing both residues with charged glutamic acid disrupts the substrate-binding pocket and interactions.

Minor point:

The first line of INTRODUCTION said that Human ABC transporter superfamily, consisting of 49 members -----. It should be 48 not 49.

Point taken. The “49” has been corrected as “48” in the revised manuscript (Page 3, line 2).

REVIEWER COMMENTS

Reviewer #1 (Remarks to the Author):

The authors have largely addressed all concerns raised. However, data on the lower activity in liposomes and nanodiscs should be included in the manuscript, at least as a supplemental figure, and briefly discussed, especially for comparison to earlier studies on ABCA4 in liposome (eg. ahn et al, JBC 2000, sun et al JBC 1999, Quazi and Molday, PNAS 2014, and others).

Reviewer #2 (Remarks to the Author):

The authors replied to all comments. However, there are still some concerns.

1. Electron density are scarcely seen in lid domain in ExFig 11a. The structure of the lid domain in Fig. 5 is not supported by the data. Therefore, the conclusion "The co-folded ECD moves as one conjoined rigid body from the nucleotide-free state to the ATP-bound state since the overall structure of the entire ECD is nearly identical in both states," at the bottom of page 12 should be revised.
2. The conclusion "After ATP binding, the ECD exhibits an obvious rotation towards the membrane, with the largest displacement for $\sim 25 \text{ \AA}$ observed at the lid region (Fig. 5d)." at the end of the result section is not supported by the data, either.
3. The angle between ECD and the nucleotide binding domain of the ATP-bound form looks different from that of other forms in ExFig 11a. This does not fit the rigid body motion proposed in this study.
4. The ATPase activity (nmol/min/mg) should be shown in ExFig. 9c like in 9b.

Response to reviewers' comments:

Reviewer #1:

The authors have largely addressed all concerns raised. However, data on the lower activity in liposomes and nanodiscs should be included in the manuscript, at least as a supplemental figure, and briefly discussed, especially for comparison to earlier studies on ABCA4 in liposome (eg. ahn et al, JBC 2000, sun et al JBC 1999, Quazi and Molday, PNAS 2014, and others).

Point taken. The activity in liposome and nanodisc were included in Extended Data Fig. 9c in the revised manuscript. And a comment “*The difference of ATPase activity in liposome between the current study and the previous ones^{39,43,45} might be caused by the different detergents used during protein purification, different lipid compositions and methods of liposome preparation, or the method of protein concentration determination.*” was added in the revised manuscript (Page 10-11).

Reviewer #2:

The authors replied to all comments. However, there are still some concerns.

1. *Electron density are scarcely seen in lid domain in ExFig 11a. The structure of the lid domain in Fig. 5 is not supported by the data. Therefore, the conclusion “The co-folded ECD moves as one conjoined rigid body from the nucleotide-free state to the ATP-bound state since the overall structure of the entire ECD is nearly identical in both states,” at the bottom of page 12 should be revised.*

We thank this reviewer for the kind suggestion. ExFig. 11a represents the sharpened high-resolution EM maps of ECD, where the relatively poor density for lid appears noisy. The EM maps were low-pass filtered to 6 Å to demonstrate the density for lid (Figure shown below). Since the electron density for lid is relatively poor, the model building was accomplished based on the homology model of lid from ABCA1. As the model for lid of ABCA4 might not be accurate, we didn't discuss about the details of lid in the manuscript. Additionally, the lid region was excluded from the analyses about the conformational changes regarding ECD in the revised manuscript. For clarification, we've added a statement “As the electron density for the lid of ECD is relatively poor (Extended Data Fig. 11a), the model for lid might not be accurate. Therefore, all the following analyses about the conformational changes regarding ECD don't include the lid region.” before the conclusion “The co-folded ECD moves as one conjoined rigid body ...” in the revised manuscript (Page 13, line 3-6).

2. *The conclusion “After ATP binding, the ECD exhibits an obvious rotation towards*

the membrane, with the largest displacement for ~25 Å observed at the lid region (Fig. 5d).” at the end of the result section is not supported by the data, either.

We thank this reviewer for the kind suggestion. The conclusion was modified to be “*After ATP binding, the ECD exhibits an obvious rotation towards the membrane, with a displacement for ~20 Å observed at the far end of the tunnel region (Fig. 5d).*” in the revised manuscript.

3. The angle between ECD and the nucleotide binding domain of the ATP-bound form looks different from that of other forms in ExFig 11a. This does not fit the rigid body motion proposed in this study.

We thank this reviewer for the critical comment. The ExFig. 11a has been adjusted to represent ECDs in the same angle as that in Fig. 5d in the revised manuscript.

4. The ATPase activity (nmol/min/mg) should be shown in ExFig. 9c like in 9b.

Point taken. The ATPase activity (nmol/min/mg) was shown for GDN purified ABCA4 in ExFig. 9d in the revised manuscript.

REVIEWERS' COMMENTS

Reviewer #2 (Remarks to the Author):

The authors answered to all the comments properly.

It will be better to include the following sentence in the rebuttal letter "Since the electron density for lid is relatively poor, the model building was accomplished based on the homology model of lid from ABCA1." in the section of structure determination (page 6).

The EM maps (low-pass filtered to 6 Å) in the rebuttal letter should be included in the extended data figures.

Response to reviewers' comments:

Reviewer #2:

The authors answered to all the comments properly.

1. It will be better to include the following sentence in the rebuttal letter "Since the electron density for lid is relatively poor, the model building was accomplished based on the homology model of lid from ABCA1." in the section of structure determination (page 6).

Point taken. The comment "Since the electron density for lid is relatively poor, the model building was accomplished based on the homology model of lid from ABCA1." was added in the revised manuscript (Page 6, line 11-13).

2. The EM maps (low-pass filtered to 6 Å) in the rebuttal letter should be included in the extended data figures.

Point taken. The EM maps (low-pass filtered to 6 Å) was included in Supplementary Fig. 11b in the revised manuscript.